# Inishell 2.0: Semantically driven automatic GUI generation for scientific models

Mathias Bavay[1], Michael Reisecker[1,3], Thomas Egger[2], and Daniela Korhammer[1]

[1]WSL Institute for Snow and Avalanche Research SLF, Flüelastrasse 11, CH-7260 Davos Dorf, Switzerland
[2]Egger Consulting GmbH, Hohenstaufengasse 7, 1010 Wien, Austria
[3]Alpine Software Michael Reisecker, Schiliftstraße 504, 5753 Saalbach, Austria

**Correspondence:** M. Bavay (bavay@slf.ch)

**Abstract.** As numerical model developers, we have experienced first hand how most users struggle with the configuration of the models, leading to numerous support requests. Such issues are usually mitigated by offering a Graphical User Interface (GUI) that flattens the learning curve. Developing a GUI, however, requires a significant investment for the model developers as well as a specific skill set. Moreover, this does not fit with the daily duties of model developers. As a consequence, when a GUI has been created – usually within a specific project and often relying on an intern – the maintenance either constitutes a major burden or is not performed. This also tends to limit the evolution of the numerical models themselves, since the model developers try to avoid having to change the GUI.

In this paper we describe an approach based on an XML description of the required numerical model configuration elements (that is, the data model of the configuration data) and a C++/Qt tool (Inishell) that populates a GUI based on this description on the fly. This makes the maintenance of the GUI very simple and enables users to easily get an up-to-date GUI for configuring the numerical model. The first version of this tool was written almost ten years ago and showed that the concept works very well for our own surface process models. A full rewrite offering a more modern interface and extended capabilities is presented in this paper.

## 1 Introduction

### 1.1 Numerical models

Numerical models can be defined as computational models designed to simulate and predict the behavior of real-world or physical systems. As illustrated in Fig. 1, given a set of input data (for example meteorological measurements) and configuration parameters (for example the simulation timestep and spatial resolutions), the numerical model will produce a set of outputs, for example snow cover and hydrological response of a catchment after simulating the physical processes leading to snow cover development and runoff generation. Numerical models are very powerful tools widely used in diverse fields, such as medicine,

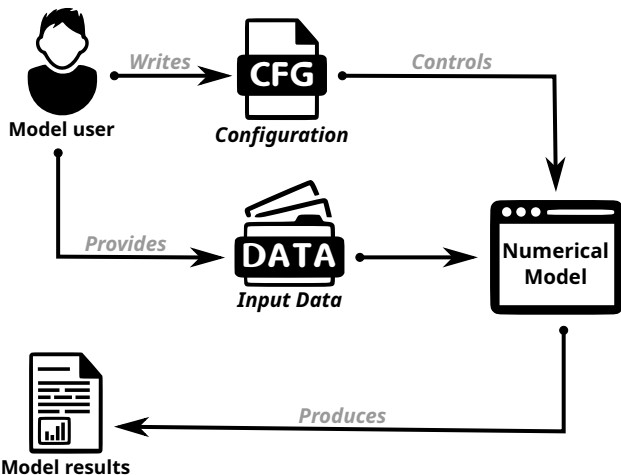

**Figure 1.** Numerical models from the user point of view

energy and environment, materials, industrial and defense as well as homeland security (Oden et al., 2006). Naturally they also see widespread use in research.

## 1.2 Configuring numerical models

When using numerical models, one of the major issues for new users is the configuration of the model (Jamieson et al., 2010; Schlögl et al., 2016; Mortezapour et al., 2020). Often the numerical models are configured by ways of multiple configuration files filled with obscure configuration parameters, making for a steep learning curve. Moreover, users tend to overlook even the best written documentation (Mendoza and Novick, 2005; Ceaparu et al., 2004) and resort to copying and tweaking example files. This is not satisfactory as it leads to under-performing simulations as well as large numbers of questions directed to the

model developers. The intrinsic complexity of configuring a numerical model may also make its end users highly dependent on direct support from the models' developers (Havens et al., 2020).

As the WSL Institute for Snow and Avalanche Research (WSL/SLF) develops and maintains the open source Snowpack model (Lehning et al., 2002) and its pre-processor MeteoIO (Bavay and Egger, 2014; Bavay et al., 2018) – two of the numerical models Inishell was originally developed for – it has first hand experience in supporting a relatively large number of end users.

A small fraction of the end users could be described as "power users" who are able to autonomously run the most complex

simulation set ups (including fully autonomous model toolchains for operational applications (Sato et al., 2004; Côté et al., 2014; Bair et al., 2020)), implement their own ideas in the model and support other users they work with. The bulk of the user base is made of researchers who want to use these models to expand their research field (Rasmus et al., 2016; Haberkorn et al., 2017; Grünewald et al., 2018; Köhler et al., 2018) and possess varying degrees of computer fluency. Finally there are practitioners who mostly do not run the models themselves but rely on outputs from model toolchains set up by somebody else (Morin et al., 2020). The first category of users usually rely on the provided online documentation and only contact the model developers for strategic questions regarding future developments or collaborations. The third category of users usually rely on their internal model experts when they have questions while the second category of users requests support from the WSL/SLF. An internal survey conducted in the Fall of 2019 at the WSL/SLF showed that user support represented around 75% of a Full Time Equivalent (FTE) when the development team represents at most 3 FTE. The requested user support would have almost been fully covered in the provided online documentation if the end users would have read it. Another short survey conducted after completion of a two weeks doctoral school snow physics and modeling course (Stockholm, Sweden) brought forth the major criticism: the user-unfriendliness of the command line and text configuration of the numerical model.

## 1.3 Graphical User Interfaces (GUI)

As reported by (Fellmann et al., 2007) for configuring Virtual Reality (VR) environments, even technical users require less support and need to spend less time looking into the documentation when a graphical user interface (GUI) is available to perform the task. Surprisingly (Voronkov et al., 2019) found that system administrators favor GUIs for configuring firewall rules (70% of the respondents) as they are perceived as good for occasional use, do not have the same long learning curve as Command Line Interfaces (CLI) and have better usability than CLI. Similarly, the configuration of numerical models can benefit from a GUI that contains inline help, input validation and also allows to predominantly show the most common settings.

Unfortunately, developing a GUI hardly fits the job description of the modelers and is very time consuming because of the potentially large number of configuration parameters: for example the Snowpack model and its pre-processor MeteoIO define more than 350 configuration keys. Developing a traditional GUI for such numerical models, where each input widget is manually laid out, would require a significant time, money and code investment (Kennard and Leaney, 2010). Although not specifically aimed at recent scientific software, (Myers and Rosson, 1992) found that on average 48% of the code in graphical applications was dedicated to the user interface, representing 50% of the development time in the implementation phase and 47% of the maintenance time. Moreover as numerical models might evolve quickly, new configuration options would frequently be added that would also require a rework of the GUI. This is hardly sustainable for small development teams and leads to either out-of-date GUIs or no GUIs at all.

The choice of tools to develop GUIs for numerical models is less than satisfying in the long term. One possibility consists of using a Rapid Application Development environment (RAD, Spreitzhofer et al. (2004)). This is easy and can appropriately be assigned to an intern or a short-term student. However this is risky in the long term since such RAD implementations are often proprietary and therefore dependent on the goodwill of its developer to maintain compatibility or even to just maintain their product, potentially forcing the model developers to perform a full rewrite of the GUI. Another possibility is using Ui-toolkits

and programming languages to develop such an interface. This requires more investment and expertise from the developer but increases the long term availability of the GUI. However, maintaining the product also requires some technical knowledge that is usually not found in model developers. At the very least, it adds a considerable workload which may have to be put off until later. Practically this means that upgrades (such as introducing new configuration options) will only happen (typically in an academic context) when another intern or student with the proper skill set can be found and funded. Delegating this task to a temporary employee however loses first-hand knowledge about these newly introduced options. Then the graphical user interface becomes a hindrance for the model itself since it prevents the fast deployment of new configuration options. Another possibility relies on Declarative User Interface Model (Da Silva, 2000) or Model-Based UI Development (Paterno, 1999), an approach that has been steadily maturing over the last several decades (Meixner et al., 2011). However the downside of this approach is that it can be highly theoretical and hard to understand and implement by designers and developers (Bogdan, 2017). Data driven GUI generation might be a middle way to reduce this complexity as it generates GUI elements from templates or from pre-existing components (Component Based Development, (Brown, 2000)) thus providing the automatic customization of the GUI for the task to be performed, based on the data that is provided (Gambino et al., 2018).

Finally, numerical models are getting more and more modular, including through coupling of existing numerical models. This leads to modules that can be used standalone or within a wider numerical model. As such, there is no centralization of the configuration data that has been provided by the end user and a centralized data model for the configuration data is not possible: the main module does not have any insight into the data model of its sub-modules. Moreover, there is usually no explicit data model for the configuration data, it is only implicitly expressed through the source code as assumptions and enforced requirements. It might also be explicitly laid out in the documentation (that must properly link to the documentation of each sub-module) or in a GUI (that must include the configuration options for all sub-modules) but they must then be kept synchronized with the implementation in the source code in order to be useful to the end user. As mentioned above, manually designed GUIs tend to lose this synchronization very quickly because of the involved workload. Therefore a solution to lighten this workload helps keeping the data model expressed in the GUI up to date and relevant for the end user.

## 1.4 Reproducible science considerations

Although the topic of reproducible science (Munafò et al., 2017) is very broad even when restricted to numerical simulations (Fitzpatrick, 2019), a GUI that is easy to keep up to date can be a technical means to help address some of the key issues. As a side effect, having an easy to maintain GUI at their disposal gives an incentive to model developers to explicitly describe the data model of the configuration data (as a GUI layout and logic), encourages them to document new features (as help text in the GUI) and to avoid hard-coded values (Cuntz et al., 2016) since making a dynamic setting be read from a user-editable configuration file is easy, quickly done and can be shown in the GUI. This contributes to making the research more reproducible (Martin, 2009) because as the numerical model code remains independent of the simulations that are performed, it does not need to be edited for each research project. It is then advised to manage the source code with a code versioning system which allows citing the specific version of the model that has been used (Fitzpatrick, 2019).

On the numerical model side, in order to allow for high modularity of the numerical models with respect to their sub-modules, the configuration data is centrally read as key/value pairs of strings into a C++ map data structure ((ISO, 1998; Meyers, 1992) similarly to a dictionary in Python, for example) but not processed any further. This data structure is then provided to each sub-module to extract and parse its supported configuration keys. Therefore the data model is delegated to the sub-modules which enforce data types, ranges, validation and dependencies between configuration keys, keeping each sub-module independent of the others (Bavay and Egger, 2014). This redundant validation (in the GUI and in the numerical model) also ensures that manually edited configuration files are supported as in such a case no GUI could perform input validation.

In order to further improve the quality of the numerical modeling work and constrain the problem, numerical model developers are strongly encouraged to rely on a single configuration file for the whole model and all its sub-modules, including as much of the pre- and post- processing as possible (but in their own sections for clarity). For example it is possible with the MeteoIO pre-processor (Bavay and Egger, 2014; Bavay et al., 2018) to start from raw data files (as they come from the data logger) and perform complex simulations such as alpine catchment hydrology. This has the advantage that a copy of the said configuration file kept together with the generated model outputs and raw inputs is then a reproducible description of the numerical simulation that has been performed (Bavay et al., 2020a). Enforcing in the GUI that the end user saves the configuration into a configuration file before running the numerical model (contrary to many interactive software often used for exploratory data analysis, (Peng, 2011)) thus makes the configuration file equivalent to some sort of a numerical simulation notebook. Moreover, having a GUI makes providing long, descriptive configuration keys painless for the end user as they can be graphically selected. This helps for the very long term reproducibility: descriptive configuration keys that provide parameters for a classical algorithm will be understandable in half a century even if the exact numerical model itself is very difficult to rerun by then.

In order to increase the possibility to rerun the exact same numerical model in the future, there are several approaches: for example as a Docker image that contains every element necessary to the reproducibility of the numerical work as laid out by (Havens et al., 2020) or as another approach by working on the numerical model and its dependencies following the Reproducible Builds[1] approach (Lamb and Zacchiroli, 2021) or ReScience efforts (Rougier et al., 2017). The former assumes that the users know how to use Docker while the long-term stability of the container file format remains to be evaluated (Rougier et al., 2017; Navarro Leija et al., 2020) and the later requires systematic tracking of all components of the toolchain. Although mature programming languages might develop some incompatibilities over time (Brunner et al., 2016), this is of concern for a very limited subset of the language and the standard tooling should be able to offer conformance to the various versions of the standard on the very long term (as is currently the case for example with C or Fortran compilers). Both approaches thus benefit from standard compliance to the programming language of the numerical model, a strongly restricted use of external dependencies (Bavay and Egger, 2014) and good software engineering practice in order to be easy enough to deploy or recompile from source in the future (including recompilation within a docker image). If these assumptions are fulfilled, the long term reproducibility should be adequate by providing the raw data, the configuration files in a text format and the exact source code (and fully documented dependencies) of the numerical tools.

---

[1]https://reproducible-builds.org/

## 2 Methodology

### 2.1 GUI Requirements

Ideally model developers would like to offer a user friendly graphical interface for configuring their numerical model that re-
quires very little initial investment and expertise and where new configuration options are quickly deployed. This configuration
interface should provide explanations of every configuration parameter, validate the user input (to avoid possible misconfigu-
rations), easily integrate new options and output the complete configuration in a standard configuration file format. This GUI
should also be able to transparently integrate the configuration options for each sub-module without requiring any duplication
of efforts and to easily support multiple numerical models.

Keeping the concept of a configuration file is important for reproducibility as well as since such models are often configured
on one system and then sent to run on some clusters to perform the heavy duty computing. This file should be manually editable
in order to allow for copy-pasting part of it between similar simulations (keeping in mind that several hundred lines might be
copied that represent hours of carefully choosing the options), to be able to modify it with text terminals through remote
sessions (as is typically the case when running on a computing cluster) or to generate at least some parts of the configuration
with scripts (for example to study the sensitivity of certain configuration parameters). It is therefore necessary to support both
the use of a GUI and of manual editing of a text file.

Finally, the users would also benefit from being able to run their numerical models from within the GUI even if this kind
of integration is only very loose (meaning that there is only very limited feedback from the numerical model to the GUI and
no interactive control of the numerical model once started) as it avoids having to open a command line terminal to run the
numerical model therefore adding unnecessary friction in the workflow.

### 2.2 General principles of Inishell

The Inishell open source software alongside numerical model design considerations is our technical answer to the previously
laid out requirements. It is written in C++ with the Qt framework[2] as a way to provide a cross-platform GUI with native look
and feel that can be reused for multiple numerical models and that is sustainable over many years. It aims to feel familiar to
the end users while considerably lowering the required skill set and time investment for the model developers and also shifting
support requests away from IT tasks to work directly concerning the models. It is open source under a GPLv3 license and
works on Linux, Microsoft Windows and Apple macOS among others.

For the user provided configuration data, as it would not be feasible to support all possible configuration file syntax choices,
a reasonable standard is enforced. The INI[3] informal standard has been chosen as it is a text format that is very easy to read and
parse with various programming languages as well as for interoperability with some existing numerical models. Its syntax is
also supported by many text editors, making manual edition convenient on multiple platforms. However its simplistic structure
can not contain enough information to define a data model and as it is created by the end user, all inputs coming from an ini file

---

[2]https://www.qt.io/
[3]https://en.wikipedia.org/wiki/INI_file

must be checked by the numerical model. The INI syntax is described in details in appendix A. On the longer term, Inishell has been written in a modular way so it is possible to develop parsers to support other, more modern configuration files standards, such as YAML.

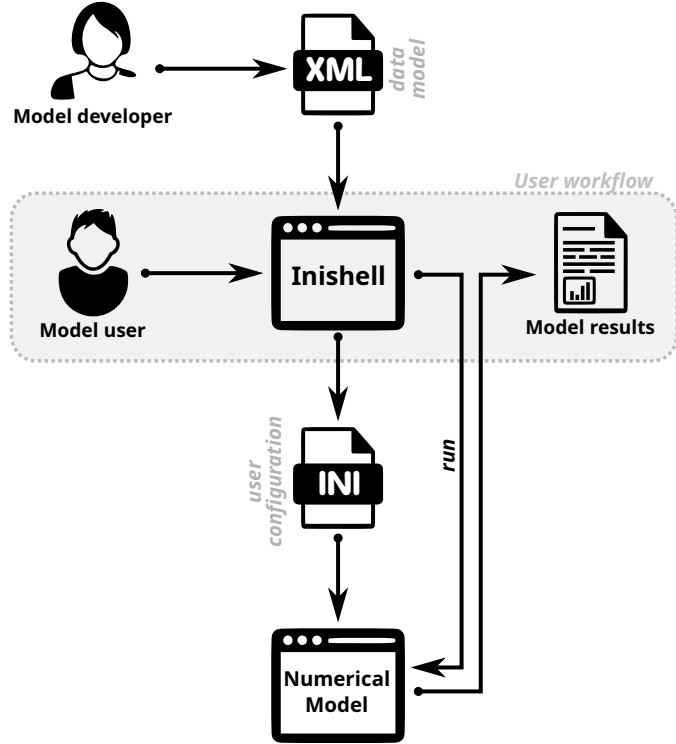

**Figure 2.** Overview of the Inishell workflows for model users and model developers

The Inishell software is a simplified derivative of Declarative User Interface Models that focuses on the data that has to be provided by the end user (high level semantic description) instead of the appearance of the GUI (low level interface attributes such as exact positioning or complex layout of the GUI). As laid out in Fig. 2, the data model of the configuration data is explicitly defined in an XML file that supports including other XML files to merge the data models of various sub-modules together. The XML syntax defines for each configuration key the key name itself, the data type of the value that should be provided by the end user (including units) as well as a help text. This basic data model of the configuration data provided by the numerical model developer is then used by the Inishell software to automatically populate its GUI that is presented to the end user to input the numerical model configuration data. Inishell enforces the user input validations at data entry time and therefore the writing of a configuration file that the numerical model can rely upon to run (more details in section 3.2). The servicing of existing GUIs and the creation of new GUIs is still decoupled from the release cycle of Inishell itself. Input

validation is supported through data types, optional range checking and by regular expressions. This is similar to the input validation provided by common JavaScript libraries such as Angular[4] or React[5]: data type, min, max, required or not, pattern.

Inishell is therefore similar to JSONForm[6] or Json-GUI (Galizia et al., 2019) and it is a higher level view of the GUI than in previous efforts such as XUL (XML User Interface Language, Goodger et al. (2001)) that is still focused on low level widgets or even UIML (User Interface Markup Language, Abrams et al. (1999)) that still keeps low level widgets as basic building blocks. It can be best compared to Atomic Design (Frost, 2016) since it is also built around a hierarchical point of view but keeping in mind that here the focus is not the entry widget type but data semantics of the data that has to be retrieved from the user.

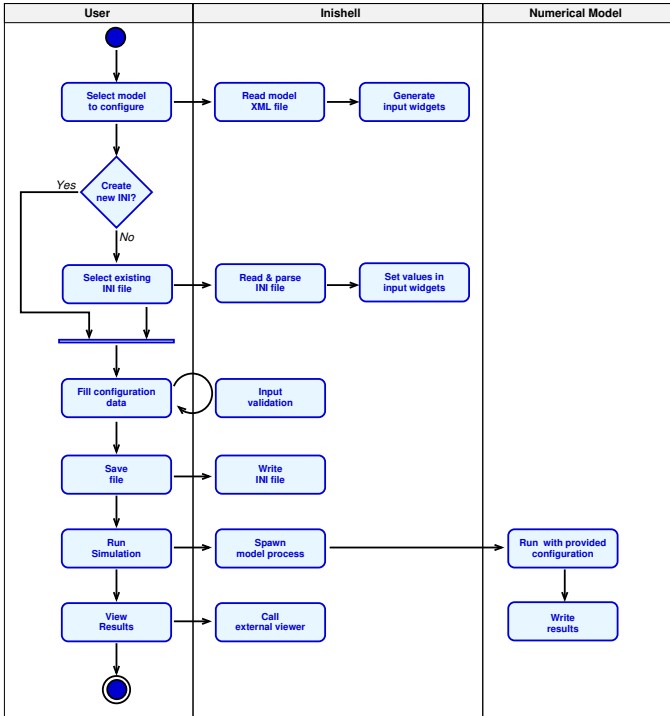

**Figure 3.** Inishell end user workflow: activity diagram

The user workflow is described in Fig. 3: the user selects from within Inishell for which numerical model to configure a simulation, which loads the numerical model's associated XML file. Inishell can then populate its GUI with input widgets that allow the user to fill configuration data or load and edit a previously written INI configuration file. Inshell performs input validation as the data is entered in order to provide the user with a quick feedback if necessary. Then the user saves the configuration file and can run the numerical model from within the GUI (which spawns a new process in the background).

---

[4]https://angular.io/api/forms/Validators
[5]https://react-hook-form.com
[6]https://jsonforms.io/

After completion of this external process, the user has the possibility to trigger the visualization of the simulation results from within the GUI, through a call to an external viewer.

## 3 Implementation

### 3.1 Overview of the interface

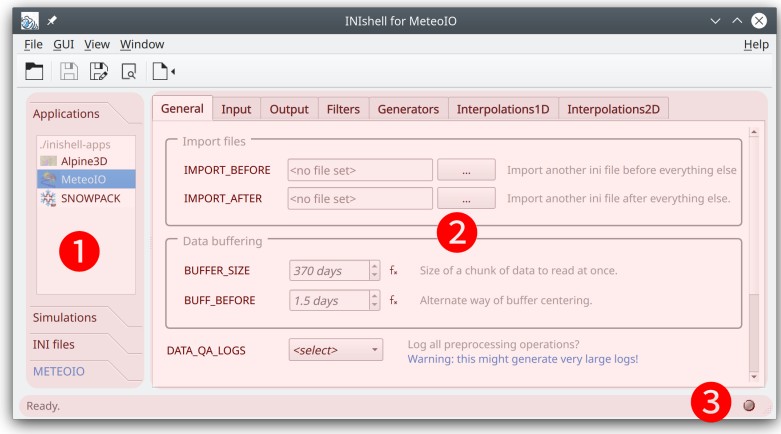

**Figure 4.** Overview of the Inishell software with the MeteoIO numerical model profile loaded (here on Linux with the light theme).

The overall interface (Fig. 4) is made of three areas, as well as a standard top menu bar. Area 1 controls the whole user workflow: In the upper panel (Applications), the users select which model they want to configure. Below (Simulations), it is possible to open a pre-configured pair of numerical model profile and configuration file (i.e. the XML file for a model associated with an INI file for the same model). One drawer lower (INI files), it is possible to open an existing configuration file. Finally, for numerical models that support it, the lowest drawer (METEOIO, shown in blue) enables running the simulation and potentially opening the simulated results.

Area 2 in Fig. 4 contains all configuration widgets for the selected numerical model profile, as provided in the XML file. It is therefore empty when starting Inishell and gets populated after loading an XML file. Once the XML file has been loaded, this area is where the end users fill their configuration parameters for the numerical model. Area 3 is a status bar that shows error messages or warnings (such as for missing mandatory configuration keys) or the status of a currently running simulation. Messages are also logged for unattended runs.

In order to further encourage end users to rely on Inishell to configure and run their simulations, a text editor is offered within Inishell under the name *Preview Editor*. It is powered by Inishell's INI format parsing and as such provides operations specifically targeted to INI files in addition to more common text editor functionality. It also keeps snapshots of the file throughout the editing process every time it is called. Hence, the Preview Editor incorporates several Inishell features into a

text editor like every user will have seen and used while still minimizing classical user errors (such as by marking unrecognized or deprecated keys as unknown with syntax highlighting).

## 3.2 General architecture

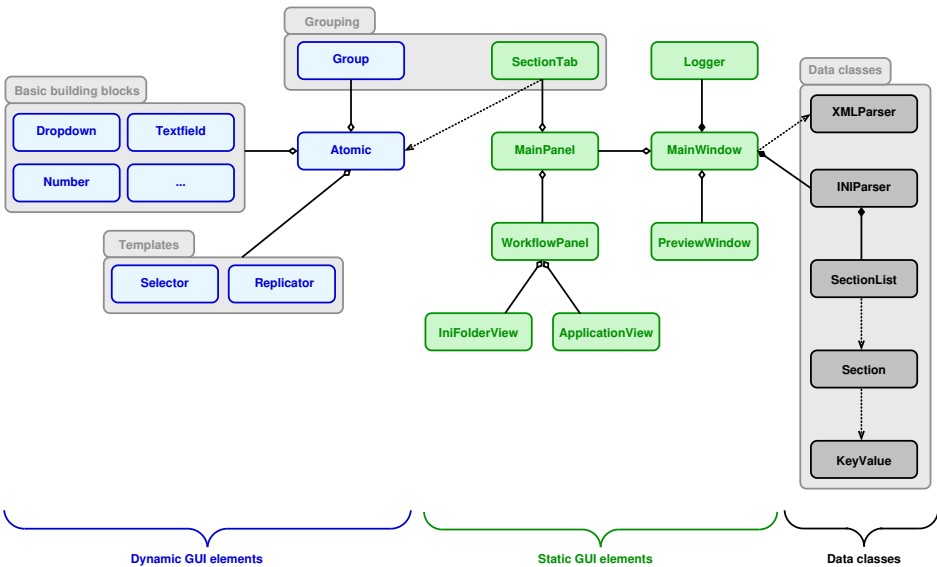

**Figure 5.** Simplified class diagram of Inishell: only the most relevant classes and relations have been represented. The classes shown in green show a set of static widgets in the GUI, the classes shown in blue are used to dynamically build widgets (based on the XML file) and the classes shown in black do not draw in the GUI.

Following the steps exposed in the activity diagram in Fig. 3, the users are first presented with only the static parts of the GUI (classes seen in green in Fig. 5). These provide the most generic features such as opening the XML file for a numerical model (*XMLParser* class), loading a specific INI file (*INIParser* class), inspecting the logs (*Logger* class) or previewing the generated INI file (*PreviewWindow* class). Once an XML file has been selected and parsed (*XMLParser* class), the *SectionTab* class that handles the area 2 in Fig. 4 recursively calls an object factory to generate widgets inheriting the *Atomic* class (that contains the properties common to all widgets) in order to populate its area of the GUI (all data that came from the XML file is then contained in the widgets). The mapping of the XML elements to GUI elements is done at this stage and is detailed later starting in section 3.3. These widgets can be input widgets or grouping widgets (by creating new tabs, the *SectionTab* class also acts as a grouping element). When the users save their configuration, the *INIParser* class is called to write out the INI file, collecting the properties from each input widget. In order to keep other user content in the INI files such as comments, line breaks (that help visually group keys together) or unsupported keys (for example belonging to another numerical model that will read its configuration from the same configuration file), all content that could not be stored in the input widgets is kept in the *INIParser* and merged back before writing out.

Inishell has a hierarchical approach both in its handling of the widgets and in the underlying architecture (where Inishell mirrors the XML structure). The most atomic elements (*atoms* in Atomic Design) are the widgets provided by the Qt toolkit. These are never exposed to the model developer, instead they are grouped into higher level elements (*molecules* in Atomic Design) by Inishell for each parameter type in the XML file (most often consisting on a label, an input widget and a help text which allows a big subset of HTML supporting quite advanced typesetting). In effect, by writing a succession of parameters belonging to sections in the XML file, the model developers set up all parameters necessary for the configuration of a module of their model, distributed over one or more tabs in the GUI that act as the next hierarchical level and are mapped to sections in the resulting INI file. These will then be grouped together under an application name that might also receive a workflow (step-by-step instructions to configure and run some model) and an icon. This is the highest hierarchical level as it matches a specific numerical model.

This hierarchical approach is simplified by relying on two modularity constructs: parameter groups and includes. Parameter groups allow giving an internal name to any group of parameters. This internal name can then be referred to later on to call this group one or multiple times. This is even more meaningful when used with the built-in inclusion system: an arbitrary number of files can be included and from them it is possible to only select the needed subset of parameters thanks to parameter groups. Several applications sharing most of the same configuration keys for any subset of their configuration can then include one file that defines all possibilities and only call the parameter groups that are relevant. In fact it is recommended to heavily rely on this system for increased modularity and decreased verbosity. In the same way models that rely on other models (e.g. in the form of libraries) can simply include this lower level model and freely extend upon it.

### 3.3 Basic building blocks

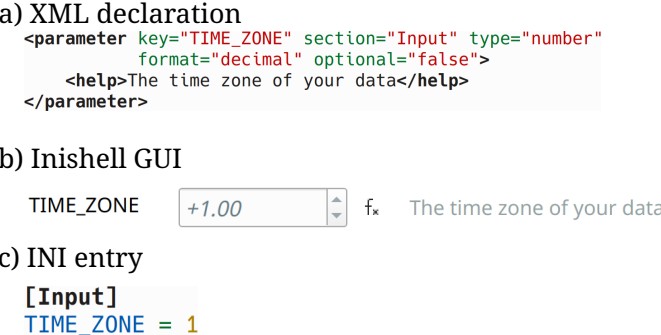

**Figure 6.** Basic building block: integer entry, as defined in the XML file, as shown in the GUI and as generated after user entry in the INI file

Inishell supports the following data types: strings, dates and times, paths to files and paths, decimal numbers, integral numbers and booleans, usually with several display options. Strings are less strictly defined as this type can accommodate free text entry or a selection among a preset list of choices (that can potentially be extended by the end user). Geographic coordinates are matched within strings through a regular expression that triggers the generation of an additional button that

shows the provided coordinates on an online map. Strings can be validated by means of a regular expression, as well as through an expression parser to make them suitable for mathematical formulas.

For each data type, Inishell generates a low level entry widget prefixed with a label that shows the matching INI configu-
ration key (or another, better suited label chosen by the model developer) and followed by a help text (that may also contain hyperlinks to a more exhaustive online documentation). Hence, Inishell manages several abstraction layers for the programmer and adequately adding and describing a model setting in the right place is now as easy as adding an XML text node with-out the need to recompile any software. Several properties for each INI configuration key can be declared. Among those, the XML property *optional* when set to *false* visually emphasizes the widget and displays a warning message when saving the file without setting it. In such a case, all the mandatory keys that have not been set by the end user will be highlighted, listed in a message, and the user can cancel saving. Manual styling of all of the used fonts is possible. Colors can be chosen freely with an RGB hexadecimal representation, but Inishell also offers a set of predefined colors with symbolic names (such as *warning*, *info* ...) which have been designed to keep good visibility if the end user changes the GUI theme, for example when using the dark theme or system wide accessibility settings.

## 3.4 Grouping elements

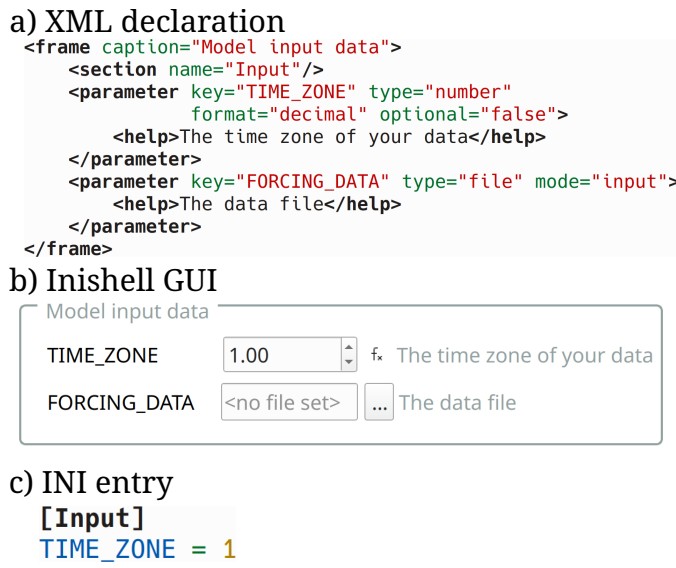

**Figure 7.** Constructs to visually grouping elements together: sections and frames

The first grouping element is matched to an INI structure: sections. It is either expressly declared in the XML or indirectly as the basic building blocks can declare which section they belong to. In the GUI, this is represented by a tab, so all INI keys belonging to a given section will have their matching widgets appear in the same tab. The end user has an overview of all the sections with the list of tabs on the top of Inishell (Fig. 4, on top of area 2).

Another grouping element is available that does not match any INI structure: frames. A frame is used to graphically group basic elements that belong together, for example a set of configuration parameters all related to the same concept in the numerical model. A frame can have its own help text which can be convenient to describe in details the feature that is configured by the keys within the frame.

## 3.5 Templates

a) XML declaration

```

    <help>File name for station number #</help>

```

b) Inishell GUI

| STATION# | + | - | File name for station number # |

| No 1: | WFJ.smet | ... |
| No 2: | DAV.smet | ... |

c) INI entry

```
[Input]
STATION1 = WFJ.smet
STATION2 = DAV.smet
```

**Figure 8.** Simple example of templates to generate as many input widgets as the user deems necessary with an auto-numbering scheme

Some fragments of the INI configuration file might have to be repeated multiple times, for example to iterate over multiple input files or over meteorological parameters. In this case, a base key is defined (for example "STATION") and multiple versions derived from this base key will be generated on-demand as requested by the user (for example by clicking on a "+" button to generate "STATION1", "STATION2" . . . ). This lets the end user provide as many variants as necessary without having to hard-code the configuration keys for each variant. In the XML file, it is handled with a system of templates where the iterators

are defined first (for example, as integral numbers or as a fixed list of strings) followed by the group of configuration keys containing a wildcard character. Inishell will then dynamically generate as many entry widgets (or groups) as asked by the end user and write all resulting INI keys in the output file.

## 3.6 Nested widgets

Some dedicated widgets offer the possibility to include more configuration options that will be shown only when a certain

choice is selected by the user (Fig. 9). This allows offering more configuration options related to a given sub-module if the

a) XML declaration

```xml

    <section name="Input"/>
    <option value="SMET">
        
        
            <option value="TRUE"/>
            <option value="FALSE"/>
        
        
        
            <option value="TRUE"/>
            <option value="FALSE"/>
        
    </option>

```

b) Inishell GUI before combobox selection

c) Inishell GUI after combobox selection

**Figure 9.** Example of nested widgets (for clarity, the help texts have been removed). Please note that the METEO key uses an alternate label and is defined as mandatory. Once it is selected as SMET, more widgets appear including the METEOPATH that is then also mandatory.

said sub-module has been enabled (for example, ticking a checkbox could show further options of the same INI section and so could the selection of specific list entries). This is a recursive process and allows for indefinite nesting.

## 3.7  Workflows

In order to allow the end user to run the numerical model from within Inishell, it is possible to declare the necessary workflow
in the XML file. This includes command line programs as well as their command line options (based on the data types that are provided by the end user), directory views (for example to open the model results directory) or opening URLs (for example to open an online viewer). As mentioned in section 2.2, running command line programs is performed by spawning a new process for the said command line program and is currently only supported on the current computer (ie. no remote execution) and does not support running through a batch scheduler (such as Sun Grid Engine or SLURM). The terminal outputs of the applications
started by Inishell are captured and shown in Inishell's main window with some basic syntax highlighting in order to highlight error messages or warnings.

a) XML declaration

```xml
<workflow>
    <section caption="DEMO">
        <element type="label" caption="Start date:"/>
        <element id="start_date" type="datetime"/>
        <element type="label" caption="INI file:"/>
        <element id="ini" type="text" default="${inifile}"/>
        <element caption="Run DEMO" type="button">
            <command>my_demo -c %ini -b %start_date</command>
        </element>
        <element type="label" caption="Visualize results:"/>
        <element id="visualize" caption="Open niViz" type="button">
            <command>setpath(%outpath, ${key:OUTPUT::PATH})</command>
            <command>openurl(https://run.niviz.org)</command>
        </element>
        <element type="label" caption="Then drag your desired
                                      output file into niViz from below:"/>
        <element type="path" id="smetpath"/>
    </section>
</workflow>
```

b) Inishell Workflow

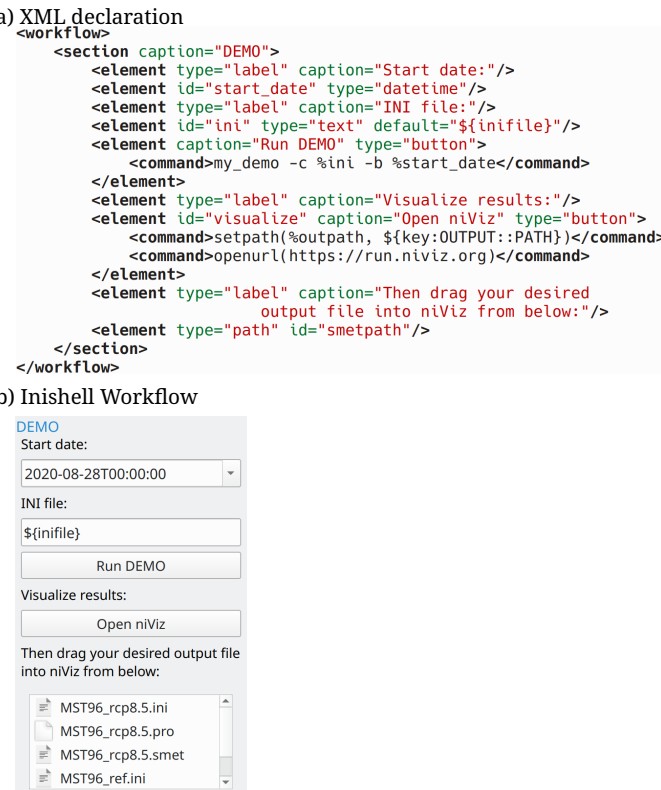

**Figure 10.** Example of a workflow panel: a few command line parameters must be provided by the user who can then run the numerical model and open a visualization application

## 3.8 Applications

Since multiple profiles of numerical models can be loaded into Inishell by opening their respective XML files, it is necessary to visually show which choices of numerical models profiles are supported and easily change between them. This is achieved by providing some meta data of the applications' properties in the XML files that define the previously described XML elements for the configuration widgets (and potentially a workflow). An application will therefore consist of a name and an icon in the applications panel, several tabs with configuration options in the main panel and often a workflow to run the application.

Combining the features listed here (choosing an application from a list, optionally auto-loading an INI file and a coupled workflow), developers can set up a list of all their models' workflows and simulations. Inishell can then handle everything from configuring the model, running it and performing maintenance work (by executing user defined system commands) – all with the click of a button within one uniform GUI and without any programming necessary.

## 4  Discussion

Although fully automatic GUI generation by Declarative User Interface Models requires complex modeling (Machado et al., 2017; Meixner et al., 2011), Inishell fully automatically generates a GUI based on a simple, high level description of the data model. This has been made possible by restricting Inishell to the narrow and simple use case of configuring scientific numerical models, contrary to more generic approaches such as (Díaz et al., 2020). In this use case, the numerical models run from a static configuration file without feedback to the GUI other than textual information (such as progression indicators, warnings and errors) and no interactive coupling – the user can not change the configuration data while the numerical model is running and the numerical model can not change its configuration data (as is typically the case when simulating over a domain defined in time and space). This has several important consequences that lead to a great reduction in complexity. First, the palette of interaction patterns is reduced (Machado et al., 2017) to Create, Read, Update and Delete (CRUD, Martin (1983)) operations. Then the reduction in application domain to a narrow scope allows reducing the required descriptive capabilities of the data model (Schaefer et al., 2006; Meixner et al., 2011) and data model description file and syntax (Galizia et al., 2019). This in turn makes the data model map to a very limited palette of input widgets (as shown in section 3.2) and the event model is even simpler (feedback stemming from input validation, hiding/showing elements based on another element's value). The focus is also not on the visual appearance of the GUI but on the data that has to be provided by the end user. Finally, the selection of supported platforms is restricted to traditional desktop computers, removing the need for one layer of abstraction (Paterno' et al., 2009). Inishell is therefore a practice-driven simplification of Declarative User Interface Models or Model Based UI Development to make this approach usable by non specialists similarly to other efforts for web-based forms such as (Galizia et al., 2019; Fardoun et al., 2018) or JsonForms.

As the Web is becoming the platform of choice for more and more complex tools, a web version of Inishell which could be integrated within a system processing the generated configuration files directly on the web certainly would have its benefits. However, a major negative feedback from some of the scientific numerical models that we develop was that the users had to open a terminal, go to the proper directory and run the numerical model from the command line. This has created many support requests and frustration from the users. Due to sandboxing and obvious security reasons, running a local executable is not permissible from a web application, which is the main reason the implementation as a native client was the preferred choice. Moreover as web technologies evolve quickly, the long term maintenance and evolution of web applications is a hurdle for research groups that must rely on external contractors for their development including trivial bug fixes. Furthermore, the interaction of a local web application with files in arbitrary locations on the system remains cumbersome.

The choice of file formats for the configuration files (currently INI) has been the result of compromises between ease of use and robustness for manual editing by the end users on one hand and expressiveness and compatibility with existing standards on the other hand. The legacy numerical models and tools have also weighted in for an easy transition to this new system and for a less risky migration. A similar project starting from scratch without any retro-compatibility issue would most probably rely on more modern and better defined standard file formats. Future developments of Inishell could benefit from supporting several output formats in order to generate configuration files for a wider range of scientific numerical models.

Since the first version of Inishell was written in the Java programming language in 2011, it is possible to draw some conclusions related to its real life impact. Overall it has worked well, allowing multiple numerical models to evolve freely without worrying about tedious redesigns of the GUI. Most of the additionally introduced configuration keys could be declared in Inishell in a matter of minutes and the possibilities offered by the XML elements recognized by Inishell have been mostly adequate. Support requests by end users of numerical models have dramatically dropped for users of Inishell since it was launched. However, some issues have been identified and addressed in the current version. First, as the Java environment is often not installed by default on personal computers anymore, it has started to cause more support requests related to the installation of Java as well as its configuration; the move to C++ is a response to this issue. Moreover, the original version of Inishell missed the possibility to run the numerical models directly from within its own interface and this has been identified as a major hindrance towards having more users rely on Inishell for their day to day simulations. This new capability has been brought through the expansion of the descriptive capabilities of the XML elements so Inishell now offers a fully self sufficient environment for configuring and running the numerical models that rely on it. This means that end users do not need to work through a combination of tools that tended to encourage them to manually tweak the configuration files (and therefore introduce errors) but find everything they need in one integrated package. This has also significantly improved the uptake of new numerical models features as end users now visually see new options in the GUI instead of having to read through many pages of documentation or detailed changelogs (similarly to what was reported for technical users in section 1.3).

The new version has since been used in complex operational simulation toolchains with completely different numerical models than it was originally developed for. Merely by adhering to the INI syntax it was possible to adequately set up the models' parameters through Inishell, document them and offer an easy to use and familiar GUI to the people running the models.

## 5  Conclusions

Scientific numerical models require a large number of configuration parameters to operate that are generally quite complex to set up. Providing a Graphical User Interface (GUI) to set up such configuration parameters improves the control that the end users have over the numerical models beyond what a standard documentation would do. However, standard GUIs are very time consuming to program for these large numbers of configuration parameters and often require a skill set that is not found in such numerical models developers. By relying on an approach derived of Declarative User Interface Models and restricting itself to the narrow use case of scientific numerical models configuration (a low complexity use case), Inishell allows model developers to quickly define in an XML file the configuration parameters that must be provided by the users along with a few properties and then generate on the fly a GUI based on these definitions. The maintenance of the GUI solely consists of editing this XML file, for example to add new configuration parameters. Ten years after the first version of Inishell has been deployed in the field, this concept has globally worked well and has been efficient both from the end users point of view and from the numerical models developers point of view. Enforcing a well defined syntax and a single configuration file has also brought added benefits such as improved reproducibility.

```
 1   [General]                            ; entering the "General" section
 2   #meteo data input settings             this whole line is a comment
 3   BUFF_CHUNK_SIZE = 370                ; this is an inline comment
 4   METEO           = SMET               ; value as string
 5   METEOPATH       = ./input/meteo      ; a path is also a string
 6   STATION1        = FLU2               ; providing two station IDs
 7   STATION2        = FIR2
 8
 9   [Filters]                            ; entering the "Filters" section
10   TA::filter1     = min_max            ; namespace for key "filter1" is "TA"
11   TA::arg1        = 240 320            ; (here short for Air Temperature)
12
13   RH::filter1     = min_max            ; another "filter1" key, but in namespace "RH"
14   RH::arg1        = 0.01 1.2
```

**Figure A1.** Syntax of the INI file with numbered lines

*Code availability.* The current version of Inishell is available from the project forge https://code.wsl.ch/snow-models/inishell under the GNU General Public License v3.0 (GPL v3) license. The exact version of Inishell presented in this paper is archived on envidat.ch (Bavay et al., 2020b).

## Appendix A: Supported INI file syntax

Although best practices have emerged that make the INI informal standard reasonably usable as a configuration file syntax, it is too loosely specified to be easily automatically generated and therefore has been defined more strictly for this work as well as extended to better suit the needs of numerical models.

The general format consists of a list of key/value pairs, delimited by an '=' sign (line 3 in Fig. A1). The values can be of type doubles, integers, boolean (*true*/*false* or 0/1) or strings. It is possible to add comments: all characters following '#' or ';' will be considered to be comments until the end of the line is reached (lines 2 & 3 in Fig. A1). The keys can be grouped by sections in order to bring more clarity and structure to the configuration file, each section being marked by a section name between square brackets (lines 1 & 9 in Fig. A1). Spaces and tabs can be used freely between words (either keys or values). Each key must appear only once per section but the same key can appear in several sections: for example a time zone information can appear in an input and an output section.

In order to keep the uniqueness of the keys in each section while allowing semantically identical keys to coexist, several extensions have been defined. A first possibility is to simply add a number after the key, making it in effect unique but clearly showing the user that all these keys participate to the same concept (lines 6 & 7 in Fig. A1). Another possibility is that a key may receive several values by providing the different values space-delimited after the equals sign (line 11 in Fig. A1). Finally, a weak concept of namespaces has been introduced: a key can be prefixed by a namespace so multiple keys belonging to different

namespaces can coexist in the same section. This makes it possible for example to declare keys for specific meteorological parameters by using the meteorological parameter abbreviation as namespace (lines 10 & 11 in Fig. A1).

*Author contributions.* M. Bavay lead the project from the beginning, contributed maintenance and development on all versions. He also wrote the bulk of the paper. M. Reisecker provided the bulk of the development of the new Inishell as well as maintenance since then and also contributed to the paper. T. Egger assisted with the implementation of the first version, helped maintain it over many years and contributed to the paper. D. Korhammer co-designed and implemented most of the original Inishell.

*Competing interests.* The authors declare that they have no conflict of interest

*Acknowledgements.* The authors are very thankful for the continued support of Charles Fierz and Michael Lehning who trusted us with our vision. Four anonymous reviewers helped to improve the paper significantly with their very insightful and constructive comments.

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
