# Peer review of "Inishell 2.0: Semantically driven automatic GUI generation for scientific models"

_Geoscientific Model Development, 2020_

## Referee Comment (RC1) · Anonymous Referee #1 · 29 Dec 2020

The paper concerns a framework aimed at the automatic generation of GUI by means of an XML description and an INI file containing structured information about the parameters having a relationship with each widget.

The manuscript is well written but I think that the system could be improved and the description as well.

Some questions:

1. There are plenty of widget toolkits. Why do you use Qt? Is this framework suitable to other widget toolkits/programming-languages?

2. Is this framework general purpose?

3. You use too much the word "model". I think that you use the word "model" also to point out the "template of the GUI layout". Moreover, the "numerical model" is an entity where the association between one widget and one parameter is accomplished (?). On the contrary, Must the association be "handmade"?

4. The information model is not defined in your work. For e.g., you could use an ERD scheme to define the structure of the information in your work. The UML class diagram could show the code structure and the relationships among the classes. (See the first suggested reference at point 8. to see how to define an information model).

5. There are two aspects in a GUI: the layout and the callback functions. I cannot see the latter aspect. The entire system seems to be a "semi-automatic form filler" because there are no functionalities to be evoked.

6. The system is not completely automatic: it seems an interface to generate another interface. Indeed, the GUI layout should be inferred from the parameters. Given a specific domain, there should be a module aimed to perform the creation of the layout on the basis of the parameters. This module should be able to create the XML related to the layout. It shouldn't be necessary an artificial intelligence approach: just a data management approach could be sufficient. For e.g., a table containing the associations: INI ENTRY station1=xxx.smet —-> ASSOCIATION TABLE station=filename —-> AUTO-MATICALLY GENERATED XML CODE: <parameter key="STATION#" type="filename" .......

7. A validation section is absent. I do not pretend a user experience study, but a section where the use of your system is easier than the "traditional" way.

8. The paper is lack of recent bibliography. The following two papers are interesting but you should include other papers recently published.

Orazio Gambino, Leonardo Rundo, Vincenzo Cannella, Salvatore Vitabile, Roberto Pirrone, A framework for data-driven adaptive GUI generation based on DI-

COM,Journal of Biomedical Informatics, Volume 88,2018,Pages 37-52,ISSN 1532-0464, https://doi.org/10.1016/j.jbi.2018.10.009.

Chunyang Chen, Ting Su, Guozhu Meng, Zhenchang Xing, and Yang Liu. 2018. From UI design image to GUI skeleton: a neural machine translator to bootstrap mobile GUI implementation. In Proceedings of the 40th International Conference on Software Engineering (ICSE '18). Association for Computing Machinery, New York, NY, USA, 665–676. DOI:https://doi.org/10.1145/3180155.3180240

Look forward to the revised version of the manuscript.

Thank you.

---

## Author Comment (AC1) · 5 Feb 2021

Answer to Anonymous Referee #1

>1. There are plenty of widget toolkits. Why do you use Qt? Is this framework suitable to other widget toolkits/programming-languages?

We have chosen Qt for its multi platform abilities that go beyond the widgets, including some low level functions that allow us not to have to care about low level platform specifics while offering a good integration within each supported platform. Moreover, it is object oriented which we found helpful for our tool and it is open source which means that are source code can be recompiled by all our users. Qt is also one of the oldest and best maintained toolkits to write generic (C++) GUIs and a de facto standard in
this domain.

>2. Is this framework general purpose?

Yes, Qt is general purpose (it offers both GUI widgets and non graphical functions to fully abstract the platform differences).

> 3. You use too much the word "model". I think that you use the word "model" also to point out the "template of the GUI layout". Moreover, the "numerical model" is an entity where the association between one widget and one parameter is accomplished (?). On the contrary, Must the association be "handmade"?

All occurrences of "model" in our manuscript refer to a numerical model, that is a computer program that simulates physical processes such as the Snowpack snow cover model, the WRF (Weather Research and Forecasting) model, the ECMWF models or any Numerical Weather Forecast model.

We have significantly rewritten the introduction and the "Principles" section because it appears that they severely lacked clarity and we have also introduced a discussion of our approach from the point of view of data models since it should be an interesting addition. A new figure has been created to show the general principles of numerical models from the user's point of view and the figure showing the principles of Inishell has been slightly expanded for clarity.

>4. The information model is not defined in your work. For e.g., you could use an ERD scheme to define the structure of the information in your work. The UML class diagram could show the code structure and the relationships among the classes. (See the first suggested reference at point 8. to see how to define an information model).

This is also a consequence of our lack of clarity: we do not define a data model in our work because Inishell consumes a data model provided as an XML file to populate a GUI where a user can enter configuration inputs (for example the spatial resolution or the timestep) for a numerical model (for example Snowpack) that will be stored as

an INI file. This INI file is then read by a numerical model (for example Snowpack) to produce some outputs. So the numerical model assumes a given data model for its configuration data that is then formerly laid out in the XML file (to be written by its developers, not by Inishell's developers). Therefore, we have the following components: 1) a numerical model written by a third party (ex. Snowpack) that can read its configuration from an INI file; 2) an XML file that describes the data model of the configuration inputs of the numerical model (to be written by a third party, for example Snowpack's developers); 3) a GUI populated by Inishell based on the provided XML file; 4) configuration inputs provided by the numerical model user by the mean of the Inishell generated GUI and stored as an INI file by Inishell.

This has now been described in the rewritten sections.

>5. There are two aspects in a GUI: the layout and the callback functions. I cannot see the latter aspect. The entire system seems to be a "semi-automatic form filler" because there are no functionalities to be evoked.

There are not really any callback functions besides implementing what has been specified in the provided XML file: the GUI widgets receive the user inputs, validate them based on the rules defined in the XML file and write them down into an INI file. In this sense, Inishell is an automatic form generator with a few auxiliary features (such as generating a preview of the INI file or loading an existing INI file). Inishell does not "process" data or produce any outputs other than the conversion of the user provided inputs in the GUI into an INI file with its syntax.

>6. The system is not completely automatic: it seems an interface to generate another interface. Indeed, the GUI layout should be inferred from the parameters. Given a specific domain, there should be a module aimed to perform the creation of the layout on the basis of the parameters. This module should be able to create the XML related to the layout. It shouldn't be necessary an artificial intelligence approach: just a data management approach could be sufficient. For e.g., a table containing the associations: INI

ENTRY station1=xxx.smet -> ASSOCIATION TABLE station=filename -> AUTOMATI-CALLY GENERATED XML CODE: <parameter key="STATION#" type="filename"

In order to remain generic, there is no domain-specific knowledge in Inishell. Each numerical model might come with its own parameter names and meaning. For example, even within a specific numerical model the same parameter name might have different meanings depending on the context: a configuration entry such as STATION1 might receive a filename or a database key or even geographic coordinates within the same numerical model, depending on the context! This means that there is not enough information to generate a relevant GUI only with the parameter names. Moreover, the INI file is quite user friendly (because of it is only very loosely structured) but it can not provide any semantic information (such as data types, ranges, regular expressions for validation or dependencies). Since the INI file will be edited or even created from scratch by the users, it is also not possible to store there any help text or other information. Therefore, the XML file contains all the semantic information for a specific numerical model (and it is often the only place where such information is formally expressed besides partly in an implicit, diffuse way within the numerical model source code, as assumptions or checks about the configuration data).

>7. A validation section is absent. I do not pretend a user experience study, but a section where the use of your system is easier than the "traditional" way.

More details have been added in the conclusion about our experience with and without Inishell from the point of view of our users and from the point of view of support requests.

>8. The paper is lack of recent bibliography. The following two papers are interesting but you should include other papers recently published. Orazio Gambino, Leonardo Rundo, Vincenzo Cannella, Salvatore Vitabile, Roberto Pirrone, A framework for data-driven adaptive GUI generation based on DI- C2 COM,Journal of Biomedical Informatics, Volume 88,2018,Pages 37-52,ISSN 1532-

0464, https://doi.org/10.1016/j.jbi.2018.10.009. Chunyang Chen, Ting Su, Guozhu Meng, Zhenchang Xing, and Yang Liu. 2018. From UI design image to GUI skeleton: a neural machine translator to bootstrap mobile GUI implementation. In Proceedings of the 40th International Conference on Software Engineering (ICSE '18). Association for Computing Machinery, New York, NY, USA, 665–676. DOI:https://doi.org/10.1145/3180155.3180240

Although there are some data driven parts in Inishell, it remains quite basic: some keys are shown or hidden depending on other keys (just acting as booleans). The second paper is quite interesting but Inishall actually does somehow the opposite: without any hints regarding how the GUI should look like, Inishell populates its GUI as some sort of a regular grid of widgets.

>From the introduction: The paper concerns a framework aimed at the automatic generation of GUI by means of an XML description and an INI file containing structured information about the parameters having a relationship with each widget.

This is what we must better describe: The XML file contains the structured information that allows populating a GUI to gather user inputs and write an unstructured INI file containing these user inputs. Inishell is therefore "only" a data entry software for configuration files with no domain specific knowledge of its own, instead relying on an XML file to provide it. The INI file is an output of Inishell (although it can re-read an existing one in order to edit it) and an input for the numerical model (such as Snowpack).
* * *
[Figure]

**Fig. 1.** Principles of numerical models from the user's point of view

**Model developer**

XML *data model*

*User workflow*

**Model user**

Inishell

**Model results**

*user configuration*

INI

*run*

**Numerical Model**

**Fig. 2.** Principle of operation of Inishell from the user's and model developer's point of view

---

## Referee Comment (RC2) · Anonymous Referee #1 · 10 Feb 2021

The data model definition is not related to the use of an XML where the information is stored. If you read the paper: Orazio Gambino, Leonardo Rundo, Vincenzo Cannella, Salvatore Vitabile, Roberto Pirrone, A framework for data-driven adaptive GUI generation based on DICOM, Journal of Biomedical Informatics, Volume 88,2018, Pages 37-52, ISSN 1532-0464, https://doi.org/10.1016/j.jbi.2018.10.009

you will find that the authors defined all the information parts (entities and relationships) and the interaction diagrams for each task of the system and the authors use an XML file to store the information. Your manuscript wasn't clear for this reason.

By the way, a manuscript with only 10 references, where 3 over 10 are auto-citations, is not permissible for a journal. Also for this reason I suggested including other works in the references, like the ones I mentioned in the previous review, even if they belong

to other domains. Substantially, there isn't a good background literature where other methodologies are applied to solve similar problems.

---

## Referee Comment (RC3) · Anonymous Referee #2 · 22 Feb 2021

It's always challenging to make the scientific models available for reuse, and one of the difficulties is to provide a clear definition of a model and its input/output parameters. The paper presents a tool developed for facilitating the generation of the configuration for models, which I think can be helpful for addressing the issue. However, I have few concerns about this paper:

1) It seems to me that the authors did not adopt the best techniques for the objective. For example, the authors adopted C++/Qt for creating the GUI, which means that the tool is dedicated to the local desktop environment. I was hoping that the tool can be Web based to allow wider and easier access. Another example is about the XML and INI. I understand that INI is more human friendly; however, a better option is YAML, which is also human friendly and better on supporting complex hierarchical structure,

which is important for describing the models.

2) The paper did not provide a clear approach on handling geospatial information, which is quite common for geoscientific models. I noticed that the paper briefly mentioned about the geographical coordinates, but did not mention other forms of geospatial information, such as polygon, raster.

3) It is important to clarify how the tool is compatible with the standards and specifics that are using by the community, such as these from the OGC. Otherwise, it will be a closed system that is hard to adopt by others.

4) The paper has been written like a technical document instead of a research article. It has been focused on presenting the specs of the tool, but not much on the justification of the approach and the scientific contributions the tool can bring to the community, especially on reusing the scientific models.

---

## Author Comment (AC2) · 1 Mar 2021

Answer to Anonymous Referee #2

> 1) It seems to me that the authors did not adopt the best techniques for the objective. For example, the authors adopted C++/Qt for creating the GUI, which means that the tool is dedicated to the local desktop environment. I was hoping that the tool can be Web based to allow wider and easier access. Another example is about the XML and INI. I understand that INI is more human friendly; however, a better option is YAML, which is also human friendly and better on supporting complex hierarchical structure, which is important for describing the models.

Regarding the choice of execution environment, the authors had an extended period

of time to reflect on this topic before starting working on the major rewrite that is presented in this paper. We considered both a web based version and a fat client version. In the end, we felt that on the long term, most probably both version will be required, although for different use cases. As we have also been leading the development of a complex web based data visualization tool (see https://niviz.org for visualization and editing of snow profiles, 121k lines of code) over many years, we have gathered experience in both kind of technologies. A web based version offers an easier access (no need to install anything) but suffers from our point of view from the following limitations: first, we had noticed that for our users, being able to run the numerical model from within the GUI is a major requirement. On of the major negative feedback to our numerical models was that the users had to open a terminal, go to the proper directory and run the numerical model from the command line. This had implications in terms of support requests (many requests were directly related to using the terminal), users frustration and acceptance of the original Inishell GUI (many users would not go back and forth between Inishell and their terminal emulator and ended up discarding the original Inishell, thus loosing the benefits of input validation, online documentation etc). Unfortunately, because of the sandboxing requirements of web technologies it is not possible to run a local program on the user's computer from a web environment. Another limitation that we saw is the relative lack of maturity of most of the client-side web technologies. A complex web based tool will be based on numerous third party libraries that so far evolve very fast and often lead to incompatibilities in a very short time frame. This means that the long term maintenance of a complex tool almost mandates full rewrites of the said tool every couple of years in order to migrate away from deprecated dependencies. This represents a major investment for an open source software that is not self funded. Finally, the programming model of Qt/C++ is much more familiar to the traditional scientific model developer than that of Javascript, meaning that low complexity changes can be implemented by a scientific model developer if need be in the fat client version while almost any change in a web based tool requires an outsourcing contract (although the idea behind Inishell is to avoid having to edit the

source code, there might always be some minor bugs and annoyances to fix).

Regarding the choice of file formats, it was also a compromise. We considered moving away from the XML files to store the configuration data data model but preferred to keep this format in order to reduce the risks associated with the rewrite that we were performing. As these files are not directly exposed to the end users, we might still move to a different format and different syntax at a later stage. But independently of the format and syntax, the main point is that the current level of expressiveness covers adequately the needs of multiple scientific numerical models. The format of the storage of the configuration data (INI files) has also been the result of a compromise. Within our field of cryosphere modeling, there is already a wide range of strategies to deal with configuration parameters, from direct source code editing (the parameters are therefore hard-coded, this happens often in models implemented in a scripting language such as Python, Matlab or Julia), INI variants (usually without even an explicit and consistent syntax definition), INI variants with some namespace structures (although some model only have one namespace that starts and ends the configuration file) and XML files. Some models relied on multiple configuration files or a mixed approach between hard-coded configuration parameters (that had to be manually edited in the source code) and some configuration files. We decided quite early in favor of an explicitly defined INI syntax in order to keep the configuration files sufficiently similar to the legacy ones so the end users would not struggle too much when porting their configurations to the new syntax. This is also the result of a compromise between the user friendliness of the syntax and its expressiveness and robustness: we always had to support direct editing of the configuration files (ie without relying on our GUI which is specially relevant until we can ensure that our GUI can cover all the needs of all our users) as well as editing through scripts (for automation such as running a large number of related simulations).

> 2) The paper did not provide a clear approach on handling geospatial information, which is quite common for geoscientific models. I noticed that the paper briefly mentioned about the geographical coordinates, but did not mention other forms of geospatial information, such as polygon, raster.

So far, all the geospatial information that we have encountered beyond simple geographical coordinates have been handled directly by the underlying numerical models as input data and not as configuration data, therefore this kind of information is not "seen" in the GUI itself besides providing input files or web services end points. As we've noticed that snippets of information (such as a few time ranges) are more obviously visible when provided as configuration data than as input data (somehow hidden within an input file), we might add more support for other forms of geospatial information in the future (but this will also have to be a compromise between the amount of information contained in the configuration file and the readability of the said file).

> 3) It is important to clarify how the tool is compatible with the standards and specifics that are using by the community, such as these from the OGC. Otherwise, it will be a closed system that is hard to adopt by others.

So far we have adopted several standards (such as date and time representations) but as Inishell is only focused on configuration data and mostly handles quite basic types, there is little overlap with standards such as those from the OGC (on the other hand, there is much more overlap with OGC standards in the underlying numerical models). In the future, depending on the needs that may arise, Inishell might gain more complex data types that would be more within the scope of standards such as OGC.

> 4) The paper has been written like a technical document instead of a research article. It has been focused on presenting the specs of the tool, but not much on the justification of the approach and the scientific contributions the tool can bring to the community, especially on reusing the scientific models.

Following the comments of reviewer #1, we have expanded the introduction section and reframed Inishell within the context of declarative User Interface Model (UIM, see (DaSilva, 2000) and (Díaz et al., 2020)). In this context, Inishell shows that within a niche application such as the configuration of scientific numerical models, it is both

easy and efficient to rely on this approach in order to provide a GUI to the end users. We have also expanded the description of our approach to include the strategy to deal with configuration data within the scientific numerical model. We hope to show to numerical mode developers the benefits this approach can bring: less support requests, low maintenance needs, high flexibility to accommodate new configuration options and better reproducibility of the model results. We have also added a discussion part to the paper in order to reflect on our practical experience from the point of view of model developers and maintainers.

[DaSilva, 2000] Da Silva, Paulo Pinheiro. "User interface declarative models and development environments: A survey." International Workshop on Design, Specification, and Verification of Interactive Systems. Springer, Berlin, Heidelberg, 2000.

[Díaz et al., 2020] Díaz, Eduardo, et al. "An empirical study of rules for mapping BPMN models to graphical user interfaces." Multimedia Tools and Applications (2020): 1-36.

---

## Author Comment (AC3) · 1 Mar 2021

Answer to referee #1

> By the way, a manuscript with only 10 references, where 3 over 10 are auto-citations, is not permissible for a journal. Also for this reason I suggested including other works in the references, like the ones I mentioned in the previous review, even if they belong to other domains. Substantially, there isn't a good background literature where other methodologies are applied to solve similar problems.

In the sections that we have rewritten we have also added references belonging to other domains. We have now reframed Inishell within the context of declarative User Interface Model (UIM). Here we see Inishell as a simplification of these more generic

works thanks to the much more limited complexity of handling the configuration data of scientific numerical models (reduced diversity of inputs, much simpler dependencies between input fields and much simpler workflows). We are grateful for the feedback that has helped us to step out of our domain and to present Inishell in a wider context that will make the paper more valuable to a broader audience.

---

## Editor Decision (ED1)

Dear authors,

The revised submission of your manuscript was first reviewed by two reviewers, who had also evaluated the original manuscript. Because their estimates of the manuscript after revisions were quite different, I decided it was needed to have the work evaluated by additional reviewers. That is why it took up quite some extra time and we ended up with 4 reviews, of which 3 have provided additional comments to be addressed. The additional reviewers have evaluated the work considering all existing review comments. The overall result is that the manuscript needs further revisions before it can be published in GMD. The comments are given below. I suggest we take on the point raised by reviewer 3 about which GMD manuscript type to assign the manuscript to at a later stage.

Kind regards

Heiko Goelzer

**Reviewer #1**
*Suggested mayor revisions.*

I appreciate the improvements to the manuscript, but the authors leave out some details.

No information model in any form is presented. Just a graphical abstract of the system is presented in the manuscript without the description of the architecture with classic software engineer representations.

Moreover, the bibliography has been updated without the suggested papers that I mentioned in the 1st round of review, especially the one regarding adaptive GUI generation based on DICOM:

Gambino O, Rundo L, Cannella V, Vitabile S, Pirrone R. A framework for data-driven adaptive GUI generation based on DICOM. J Biomed Inform. 2018 Dec;88:37-52. doi: 10.1016/j.jbi.2018.10.009. Epub 2018 Nov 9. PMID: 30419365.

where the authors can learn how to depict a good software design.

Look forward to the revised version of the manuscript.

**Reviewer #2**
*Believes that the authors have addressed the comments from the previous review and has recommended to accept the manuscript for publication.*

No additional comments.

**Reviewer #3**

*Suggested minor revisions.*

In Inishell 2.0, the authors present a tool that converts an XML-based description of a numerical model configuration structure into a Qt based GUI for use on a desktop operating system. Specifically, this tool is designed to facilitate the creation of INI-based configuration files used by numerical models.

The two reviewers' comments appear to fall into three main concerns. I give my summary of the reviewers' comments in addition to my thoughts.

1. The original manuscript and indeed the manuscript paper read more like a technical report rather than a science paper.

I am of two minds on this. On one hand I am missing a strong scientific contribution by this tool – what new science can be done with it? There is no new testable hypothesis and no new method codified in a tool. Rather, it is a tool that facilitates configuring existing models by translating a series of setting options into a layout the authors deem suitable. However, I believe such a contribution falls within the GMD submission guidelines for "development and technical papers, describing developments […] or technical aspects of running models such as the reproducibility of results." There is a significant push in the earth science community for reproducible science, including the configuration of numerical models. On line 80 the authors note "This has the advantage that a copy of the said configuration file is then a reproducible description of the numerical simulation that has been performed […] keeping a changelog of the configuration files […]" which cleanly links with the idea of reproducible science and codifying the steps taken during the workflow of setting up a model. However, this point is never fully described. How are the changelogs produced? Where are they stored? How does the user interact with them?

Throughout the revised manuscript there are anecdotal comments that lack sufficient context related to end-users interacting with the models and configurations. I am familiar with this groups SNOWPACK model and their ongoing work with practitioners in the avalanche hazard forecasting community. Thus, when they make statements, for example, that users struggling to use the command line terminal, it is unclear who had this issue – are these practitioners or scientific users?

I therefore think that the authors need a bit of polish to pull together the context they added in revision two to better link the concepts of reproducible science, end-users who are domain experts but not model experts, and what types of GUI features are required to support these use cases. Having a GUI that facilitates reproducible research and ensures domain experts have access to appropriate models seems to be a useful contribution. I believe that citations from the reproducible science literature to support the GUI decisions would address the reviewer's concerns [0,1]. I understand [1] to want a further description of what I believe the authors want to address: "how do we help domain experts who aren't modellers use our tools in a reproducible way." At the moment the synthesis that the authors added in revision 2 to address this feels tacked-on.

[0] "A validation section is absent. I do not pretend a user experience study, but a section where the use of your system is easier than the "traditional" way."

[1] "Substantially, there isn't a good background literature where other methodologies are applied to solve similar problems."

**2. Data model and GUI creation**

There is a substantial body of research in the Human-Computer Interaction (HCI) literature on how GUIs are designed, A/B tests to determine optimal layouts, etc. The authors note that "[t]he focus is also not on the visual appearance of the GUI but on the data that has to be provided by the end user." Certainly, there must be some method to how the UI is laid out. Is there a novel way in doing so to optimize interaction with numerical models? This links with [0] but also [1] where presenting a UI in a manner to help domain experts interact with models requires some thought in how the UI is designed.

I believe it is to this point that the reviewer #1, in 339-RC2.pdf who recommended doi: 10.1016/j.jbi.2018.10.009, is reacting. It seems the authors attempted to address this in Table 1, however it is not clear to me how this table is supposed to be read – top to bottom? Left to right? I believe that the authors should tighten section 3.2 to make it more clear how the various levels in the hierarchical design works and exactly how the XML options are mapped to these internal data models for generating the UI. I would like a more detailed description of how basic range checking and regex checking are done – is it as the user types or when the INI file is created? Lastly, and this does also relate to point 3 (generalization) below, is that describing the internal data model and class relationship as suggested by the reviewer would help demonstrate applicability to a wider gamut of toolsets. Most models do not use INI configuration files. Thus, how easily can a user swap the INI writer for a, for example, a JSON writer? These internal data model descriptions and class layout diagrams would be helpful to understand this.

I downloaded and compiled IniShell. In general, it looks nice. However, there are non-standard UI decisions that, as a first-time user, were sufficiently quirky and non-standard that I was confused. I had to check the mouse-hover tooltip to understand what it did. For example, the save icon is not the standard floppy-disk icon. The open icon looks like the standard 'new file' icon instead of the more standard "open folder". The preview icon is a printer (I assumed this printed?). In the left-hand side panel, the example INI files are listed twice (seemingly). These types of GUI design decisions and departures from well-known paradigms have a 'scientist UI' feel to it versus "lovingly hand-crafted" by a UI expert feel. It is my opinion "focus is also not on the visual appearance of the GUI" requires more justification.

**3. Generalization of the tool**

Reviewer 1 and 2 both question the general applicability of the tool including the inclusion of geospatial data. The authors stress in the manuscript that although designed for their models, it is quite generalizable. Given IniShell is a "semi-automatic form filler" there is seemingly limited tight binding to the underlying numerical models. However, the authors

do note that the INI standard "[…] therefore has been defined more strictly for this work as well as extended […]." It is unclear to me how easily it would be to implement this in another model or if INI conversion tools can handle this. I would like to see an explicit table of what was extended to clearly explain what would need to be added to a downstream consuming model to take advantage.

Regarding inclusion of geospatial data, I found the authors response satisfactory. Setting configuration keys that point to the input data is likely sufficient as detailed data validation is likely best in the pre-processing step or during model initialization.

Other notes:
I believe the authors have well addressed RC1 and RC2 questions regarding the use of Qt. In my view Qt is absolutely a reasonable choice for IniShell. Qt has been in active development for many years, has a mature community, open source, &c. Any of the big cross-platform widget toolkits would be appropriate for this work and I think Qt is perfectly valid. I agree with the authors that a web-based tool provides limited value for a scientific computing tool.

I am curious if the authors have thought about an approach to automatically generate the XML file from the model source code (also noted by the reviewer). The XML file still represents a non-zero maintenance burden and could end up out of sync of the source code despite all efforts. Code rot happens to the best of us.

Summary

Based on my review of the manuscript as well as the two other reviewers' comments, I believe that this manuscript would make a contribution to GMD under "development and technical papers, describing developments such as new parameterizations or technical aspects of running models such as the reproducibility of results." However, I think there is room for improvement in tightening the manuscript. Specifically, to: 1) improve the literature summary/gap and discussion to position this manuscript around reproducible research and expert-user/non-modeller usage to clearly meet GMD contribution guidelines; 2) more clearly articulate how the internal IniShell data model translates the input XML file into a GUI and could allow for generalization to different configuration formats; and 3) to clearly articulate the INI spec extension to ensure generalizability.

I would suggest acceptance with moderate revisions.

**Reviewer #4**
*Suggested additional revisions that mainly address issues in structure and clarity from a design perspective.*

This is an interesting paper that addresses a problem that deserves attention. The paper describes a solution to the problem, but fails to clearly identify the problem and the scope of the paper. Much relevant information comes implicitly throughout the paper, but the

paper as such would benefit from a clearer structure focusing on the problem/use case, methodology/approach/architecture, implementation, discussion and conclusion/summary. As it reads now it is difficult to follow the authors through this process. The impact of the paper would be greater if this is clarified.

In the methodology/approach/architecture, usage of interaction and sequence diagrams would be beneficial along with a clear description of the data model supporting this interaction. The illustrations that are provided are nice, but needs more support.

Specific comments

Section 1: I would like to see the introduction split to a more general background on generic GUI and a separate section on problem description which also explains the nature of numerical models and how they are configured and operated in more detail. As it is now, the sub section on requirements appears to lack justification. Additional references on the problem description would be useful.

Section 1.2 – last paragraph: this is a very bold statement which I can't see is justified in the current text. Either more references or a better problem description and analysis would be necessary.

Section 2: Should be renamed to methodology or similar preceeded by a proper problem description and justification of selected approach.

Section 2: In order to justify inishell I miss a more thorough problem description and analysis of potential solutions. There are good reasons to choose inishell, but it doesn't mean that there aren't alternatives that appears more modern. Referring to INI as the informal standard is not justified, references are needed or a more thorough analysis of approaches.

Section 2.1 – paragraph 7: Referring to the XML file as the data model for the configuration files won't do. A more generic approach would be necessary using e.g. UML or similar, representing the various sections identified in the GUI presented later.

Section 2.2 – first paragraph: What is meant by "...numerical model profile and specific configuration profile...". I am not sure a "specific configuration file" is the same as "a model with loaded settings ready to run". A better explanation is required.

Section 2.2. figure 3: The GUI representation is jumped without a proper model describing the purpose of the different sections in the GUI. A logical workflow, potentially represented through an interaction diagram or similar would be useful to understand the context.

Section 2.2 – first paragraph: To my knowledge MeteoIO isn't a numerical model but rather a pre- processor of input data. Clarification would be welcome.

Section 3: As far as I can see both section 2 and 3 are implementation. A better separation of content would be welcome. Again referring to INI as the informal standard for configuration files has to be justified. In general I would claim that YAML is at least as widespread as INI, but there can be differences between communities that for legacy reasons stick with INI. References would help in this part.

Section 3.2: I am sure I would agree on this being the architecture. As mentioned earlier, to simplify reading a general introduction followed by a proper problem description and analysis, then a section on methodology or approach that contains the architecture, leading up to the implementation and a discussion of this would simplify the reading of the paper. As sections 2 and 3 are somewhat interleaved and not clearly defined scope wise a restructuring would be welcome.

Section 3.3: Not sure I understand what is meant by semantic names, seems repetitive.

Section 3.7: The work flow description would benefit from interaction or sequence diagrams. It is not clear if GUI and model has to run in the same environment or if remote control is possible, This would be interesting to know. Also is this more than a configuration system? Is it also a execution environment?

Section 3.8: I can't see that a numerical model (nor multiple) is loaded into INIShell. Rephrasing would be welcome to clarify what a numerical model is.

Section 4 – second paragraph: For the time being running numerical models is commonly done in HPC than cloud environments, but could depend on the community and type of model. Referring to the cloud without addressing HPC would be oversimplifying the complexity of numerical models.

Also referring to my comment on section 3.7, a clarification of execution environment would be beneficial up front, not indirectly addressed in discussion.

Section 4 – third paragraph: file formats for what?

Section 5: I can't see that the statement that a GUI gives better control over the configuration than a text based solution. Justification would be required, also in the discussion. And the justification should also refer back to the problem description that is the foundation for the paper.

---

## Author Response (AR2)

**Answer to Anonymous Referee #1**

*>No information model in any form is presented. Just a graphical abstract of the system is presented in the manuscript without the description of the architecture with classic software engineer representations.*

Two UML diagrams have been added: an activity diagram to show both the expected user workflow and the actions that are triggered within Inishell and a simplified class diagram to give an overview of how Inishell is organized internally and to show how the internal organization reflects the individual graphical elements that are presented later in the same section.

*>Moreover, the bibliography has been updated without the suggested papers that I mentioned in the 1st round of review, especially the one regarding adaptive GUI generation based on DICOM*

This reference has been added.

**Answer to Anonymous Referee #2**

No further answer necessary

**Answer to Anonymous Referee #3**

*>There is a significant push in the earth science community for reproducible science, including the configuration of numerical models. […] which cleanly links with the idea of reproducible science and codifying the steps taken during the workflow of setting up a model. However, this point is never fully described. How are the changelogs produced? Where are they stored? How does the user interact with them?*

There is no special mechanism in Inishell to produce or manipulate changelogs, but by providing benefits to the end user as well as to the developers that as a side effect promote good practice regarding reproducibility, Inishell can be part of a wider system that encourages all actors to produce reproducible numerical simulations. A new section on this topic with new bibliographic references has been added.

*>Throughout the revised manuscript there are anecdotal comments that lack sufficient context related to end-users interacting with the models and configurations. I am familiar with this groups SNOWPACK model and their ongoing work with practitioners in the avalanche hazard forecasting community. Thus, when they make statements, for example, that users struggling to use the command line terminal, it is unclear who had this issue – are these practitioners or scientific users?*

The authors have tried to clarify these points. Basically, both kinds of end users are often not very technically inclined and therefore struggle with the command line terminal. This is most probably the result of a feedback loop: as a model get easier to use, they attract users who might have previously been afraid of the complexity of the model and therefore bring even more usability requests and requirements.

*>I therefore think that the authors need a bit of polish to pull together the context they added in revision two to better link the concepts of reproducible science, end-users who are domain experts but not model experts, and what types of GUI features are required to support these use cases.*

The authors have added much more content on these topics.

*> Having a GUI that facilitates reproducible research and ensures domain experts have access to appropriate models seems to be a useful contribution.*

These points come as side effects of having an up-to-date GUI for the models. The main goal of Inishell is not to facilitate reproducible research nor to provide appropriate models to domain experts, but to provide a GUI to models that previously did not have one at all.

*>[0] "A validation section is absent. I do not pretend a user experience study, but a section where the use of your system is easier than the "traditional" way."*

More feedback from our own experience with user support has been added. A few references have been added. Basically, this comes down to numerical model with GUI versus numerical model without any GUI at all.

*>[1] "Substantially, there isn't a good background literature where other methodologies are applied to solve similar problems."*

The literature that was already cited in the second version of the manuscript does cover this: there are two well known alternatives in order to provide a GUI for a software: an ad-hoc GUI (as carefully hand-designed) or a system to generate a GUI from an abstract description of the GUI (Declarative User Interface Model). None of these approaches is suitable for numerical models developed by small teams (between 1 FTE and less than 2-3 FTE, models such as the GeoTop hydrological model by Endrizzi et al.[1], the AMUNDSEN alpine snow cover model by Strasser et al.[2], the CROCUS snow cover model by Vionnet et al.[3], the radiation transfer model SMRT by Picard et al.[4], as well as the snow cover and snow hydrology models developed or maintained by the authors such as Snowpack, Alpine3D, AlpineFlow and MeteoIO). The authors have deliberately chosen not to refer to specific models in the manuscript in order to keep it generic enough (models outside of the cryospheric sciences that are developed by small teams have the same shortage of manpower).

*>There is a substantial body of research in the Human-Computer Interaction (HCI) literature on how GUIs are designed, A/B tests to determine optimal layouts, etc. The authors note that "[t]he focus is also not on the visual appearance of the GUI but on the data that has to be provided by the end user." Certainly, there must be some method to how the UI is laid out. Is there a novel way in doing so to optimize interaction with numerical models? This links with [0] but also [1] where presenting a UI in a manner to help domain experts interact with models requires some thought in how the UI is designed.*

The core concept of Inishell is to provide a GUI with very low investment for any model that had none (not only for the Snowpack model but any other arbitrary model that could read its configuration from an INI file. Actually, some other, unrelated models have implemented their GUI in Inishell). Therefore no effort has been made in order to optimize the interaction with the numerical model or the GUI layout as this would be very much too specific for one model or would require much greater complexity (this is feasible with Declarative User Interface Model but comes at the expense of tremendously higher complexity and skill thus making it impractical for the numerical models that Inishell wants to help). The new material in the manuscript should clarify the

1   Endrizzi, S., Gruber, S., Dall'Amico, M., & Rigon, R. (2014). GEOtop 2.0: sim-872 ulating the combined energy and water balance at and below the land surface 873 accounting for soil freezing, snow cover and terrain effects, Geoscientific Model 874 Development, 7 (6), 2831–2857, doi: 10.5194.
2   Strasser, U., Bernhardt, M., Weber, M., Liston, G. E., & Mauser, W. (2008). Is snow sublimation important in the alpine water balance?. The Cryosphere, *2*(1), 53-66.
3   Or the more recent Vionnet, V., Brun, E., Morin, S., Boone, A., Faroux, S., Le Moigne, P., Martin, E., and Willemet, J.-M.: The detailed snowpack scheme Crocus and its implementation in SURFEX v7.2, Geosci. Model Dev., 5, 773–791, https://doi.org/10.5194/gmd-5-773-2012, 2012.
4   Picard, G., Brucker, L., Fily, M., Gallée, H., and Krinner, G.: Modeling time series of microwave brightness temperature in Antarctica, J. Glaciol., 55, 537–551, 2009.

basic building blocks that are used by Inishell which then most of the time simply stacks in vertical grid the (label, input widget, help text) triplet. This is not fancy but remains very simple for the model developer to handle and is still a welcomed improvement over the command line interface (that is often problematic for the end users) and the documentation that is not read by the end users (see references in the introduction).

> *It seems the authors attempted to address this in Table 1, however it is not clear to me how this table is supposed to be read – top to bottom? Left to right? I believe that the authors should tighten section 3.2 to make it more clear how the various levels in the hierarchical design works and exactly how the XML options are mapped to these internal data models for generating the UI.*
The table was not really useful so it has been removed. Instead, two new UML diagrams should better show how Inishell works alongside with new text. The general principle of mapping XML elements to GUI elements is explained in the rewritten "General architecture" section and the following sections.

> *I would like a more detailed description of how basic range checking and regex checking are done – is it as the user types or when the INI file is created?*

This is now explicitly described in the new "Reproducible science considerations" section as these checks must be done twice: both in the GUI and in the numerical model itself (so the numerical model can still validate its configuration parameters even when provided by users who do not rely on Inishell to create/edit their configuration files. It still makes sense to perform it anyway in the GUI as this allows the GUI users to quickly spot obvious configuration errors without having to run through the numerical model).

> *Lastly, and this does also relate to point 3 (generalization) below, is that describing the internal data model and class relationship as suggested by the reviewer would help demonstrate applicability to a wider gamut of toolsets. Most models do not use INI configuration files. Thus, how easily can a user swap the INI writer for a, for example, a JSON writer? These internal data model descriptions and class layout diagrams would be helpful to understand this.*

This has been clarified by providing a class diagram and a description of the class diagram, including where the data is stored (as key/value pairs). Basically, any format that is based on key/value pairs with some sort of name spaces could be easily implemented. Obviously, instead of calling the data storage SectionList through the INIParser, it would make sense to call it through a generic interface class. As a side note, the syntax highlighting in the PreviewEditor would also have to be expanded in order to cover a new syntax (practically, creating another instance with different rules as well as a mechanism to determine which instance should be used based on the configuration file format that has to be generated).

> *I downloaded and compiled IniShell. In general, it looks nice. However, there are non-standard UI decisions that, as a first-time user, were sufficiently quirky and non-standard that I was confused. I had to check the mouse-hover tooltip to understand what it did. For example, the save icon is not the standard floppy-disk icon. The open icon looks like the standard 'new file' icon instead of the more standard "open folder". The preview icon is a printer (I assumed this printed?).*

First, thank you for your involvement in this review! The handling of icons is a little special: it is based on the XDG standard for icons themes (https://specifications.freedesktop.org/icon-naming-spec/icon-naming-spec-latest.html). None of the icons have been designed by ourselves nor are they selected by ourselves or referenced by filename in the source code. Instead, the source code calls standard names that explicitly describe the action or place that the icons should represent (for example, "document-save-as") and then the underlying Desktop Environment should select from its

currently enabled icon theme which icon to provide. This visually integrates Inishell better into the user's environment as the icons are consistent with the rest of the environment (so light or black themes are supported, 4K screens should receive the proper icons, etc) while being fully transparent to Inishell (and sparing us graphical design).

However, the XDG standard is not supported on Windows or MacOS. Therefore, a fallback had to be implemented. This consists of providing pre-packaged icons if the call to get the icons from the system failed. For better visual integration, two fallback icon themes have been selected (both XDG compliant), one for Windows and one for MacOS. Another requirement for these fallback themes is the license: they must be redistributable (so the native icons on each platform could not be used). We therefore selected Open Source icon themes that are popular and usually well considered. The icon theme for MacOS replaced the legacy floppy icon for file save operations by another visual metaphor. This has turned out not to be so understandable, and after looking for standard ways to represent "save file" on MacOS and finding nothing more or less standardized (only a few non-native applications provide a "save file" icon in a theme that definitely looks non-native), we have now reverted to using the same icon theme for both MacOS and Windows (this is an icon theme that uses the floppy metaphor).  However, on the longer term the floppy metaphor will loose its relevance...

>*In the left-hand side panel, the example INI files are listed twice (seemingly). These types of GUI design decisions and departures from well-known paradigms have a 'scientist UI' feel to it versus "lovingly hand-crafted" by a UI expert feel. It is my opinion "focus is also not on the visual appearance of the GUI" requires more justification.*

The XML files are listed twice because Inishell looks for XML files in multiple locations. Inishell must support multiple versions of the XML files, matching multiple version of any given model (for example, a stable release and a development version, having small differences as new modules get added or new configuration parameters). In order to help the user select which one to use, the XML files are grouped together by the paths where they have been found and the path itself is shown above each group. On mouse-over, the full path and filename are given in a tooltip.

Moreover, not being able to show any XML files (or not showing the ones that the end user expects to see) at start is a much more serious issue for the end user than showing too many. Since we want to support both users of prepackaged versions of Inishell and users recompiling Inishell themselves, we need to include many search locations (this is made even worse by the fact that each compiler tends to setup a different directory structure where it builds the binaries, which changes the relative path between the executable and its XML files and we don't want to impose a specific compiler on the users, so we need to support all the build structures variations). By default on MacOS, because the example XML files might or might not be part of the build directory (depending on which build target has been called and if the packaging could be performed or not), the example XML files might appear once (only the XML files present in the source directory) or twice (adding a copy of the XML files as resources in the .app directory structure that can be packaged into a dmg or just copied to another location while deleting the source directory or copied to another computer altogether). Of course, users relying on prepackaged binaries won't see this duplication of XML files as they won't have the source directory.

Detecting and avoiding the duplication mentioned in this comment could be avoided but at the cost of increased complexity in the build system (more detection of the various paths in use by the compiler, more exchange with the c++ implementation to skip some of these paths after checking them against the compiler-specific structure in the build directory) or by limiting support for source compilation by end users (our experience with Snowpack, MeteoIO and Alpine3D also show that supporting a relatively polished experience both using precompiled binaries and source code

compilation is a challenge with respect to resources and paths. Restricting ourselves to one or the other would make everything much simpler).

> *It is unclear to me how easily it would be to implement this in another model or if INI conversion tools can handle this. I would like to see an explicit table of what was extended to clearly explain what would need to be added to a downstream consuming model to take advantage.*

The extensions that we have implemented are now more explicitly defined in the "Supported INI file syntax" appendix. This section has been moved to an appendix in order to prevent interrupting the flow in the main document while still documenting what is supported. The other models that we know of that rely on an INI kind of syntax don't rely on these extensions and therefore don't need to change anything in order to be supported by Inishell. If a consuming model would come with its own extensions, then either this would have to be implemented in Inishell or converted to one of the extensions supported by Inishell in the consuming model itself.

> *1) improve the literature summary/gap and discussion to position this manuscript around reproducible research and expert-user/non-modeller usage to clearly meet GMD contribution guidelines;*

A new section on the topic of reproducible research has been created with some associated literature. The expert-user/non-modeller topic is now discussed in the introduction (as we saw little difference between the two groups besides a few individuals).

>*2) more clearly articulate how the internal IniShell data model translates the input XML file into a GUI and could allow for generalization to different configuration formats;*

The rewritten "General architecture" has been expanded to cover this topic alongside a new UML (simplified) class diagram that is also commented along these lines.

> *and 3) to clearly articulate the INI spec extension to ensure generalizability.*

This is covered in the new "Supported INI file syntax" appendix.

**Answer to Anonymous Referee #4**
>*The paper describes a solution to the problem, but fails to clearly identify the problem and the scope of the paper. Much relevant information comes implicitly throughout the paper, but the paper as such would benefit from a clearer structure focusing on the problem/use case, methodology/approach/architecture, implementation, discussion and conclusion/summary. As it reads now it is difficult to follow the authors through this process. The impact of the paper would be greater if this is clarified. In the methodology/approach/architecture, usage of interaction and sequence diagrams would be beneficial along with a clear description of the data model supporting this interaction. The illustrations that are provided are nice, but needs more support.*

The paper has been restructured in order to bring more clarity: the introduction now has a new subsection "Graphical User Interfaces", a "methodology" section contains a "requirements" subsection (with an activity diagram) as well as a "General principles of Inishell" section and a "Reproducible science" subsection. A new "Implementation" section contains the overview of the interface, a description of the general implementation (with a class diagram and the description of how the data that is parsed leads to the creation of widgets in the application) followed by a mode detailed description of the XML / widgets / INI mappings grouped by types of elements.

*>Section 1: I would like to see the introduction split to a more general background on generic GUI and a separate section on problem description which also explains the nature of numerical models and how they are configured and operated in more detail. As it is now, the sub section on requirements appears to lack justification. Additional references on the problem description would be useful.*

This split has been implemented in the introduction section (see above). More details about the category of users are also given alongside many more references, including on the benefits and shortcomings of GUI and some examples of numerical models users.

*>Section 1.2 – last paragraph: this is a very bold statement which I can't see is justified in the current text. Either more references or a better problem description and analysis would be necessary.*

This has been rephrased and is now part of the new "Reproducible Science" section. This also contains new references.

*>Section 2: Should be renamed to methodology or similar preceeded by a proper problem description and justification of selected approach.*

This has been done. The justification of the selected approach is found in section 1.3.

*>Section 2: In order to justify inishell I miss a more thorough problem description and analysis of potential solutions. There are good reasons to choose inishell, but it doesn't mean that there aren't alternatives that appears more modern.*

The alternatives are discussed in section 1.3 "Graphical User Interfaces (GUI)". Basically, there are traditional, manually designed GUI that involve a significant time (and skill) investment or automatically generated GUI that require very deep knowledge (as well as high complexity). Inishell aims at being a middle way that remains easy yet requires very little time investment (the limitations being that the GUI is quite primitive).

*>Referring to INI as the informal standard is not justified, references are needed or a more thorough analysis of approaches.*

We have slightly rephrased this sentence: INI is not "the" standard, but one standard although it is informal (i.e. not formally defined). It is used (often as derivative forms such as under the name "namelist") for multiple numerical models in our community, such as the Crocus environment for weather forecasting at MeteoFrance (cited above), Amundsen (cited above although a new version is coming that now uses YAML),  GeoTop (cited above), WRF (Skamarock et al.[5]), our models Alpine3D, Snowpack, MeteoIO, AlpineFlow among others.

*>Section 2.1 – paragraph 7: Referring to the XML file as the data model for the configuration files won't do. A more generic approach would be necessary using e.g. UML or similar, representing the various sections identified in the GUI presented later.*

A new class diagram has been created and commented to show how the GUI is built (section 3.2, figure 5).
* * *
5    Skamarock, W. C., Klemp, J. B., Dudhia, J., Gill, D. O., Barker, D. M., Wang, W., & Powers, J. G. (2005). A description of the advanced research WRF version 2. National Center For Atmospheric Research Boulder Co Mesoscale and Microscale Meteorology Div.

*>Section 2.2 – first paragraph: What is meant by "...numerical model profile and specific configuration profile...". I am not sure a "specific configuration file" is the same as "a model with loaded settings ready to run". A better explanation is required.*

This has been rephrased in order to be clearer.

*>Section 2.2. figure 3: The GUI representation is jumped without a proper model describing the purpose of the different sections in the GUI. A logical workflow, potentially represented through an interaction diagram or similar would be useful to understand the context.*

This figure (now figure 4) sits in section 3.1 that describes it in details (more details have been added as well as linked to the new UML activity diagram that is now figure 3). As it is a floating figure, it can be come just before the descriptive text, in the middle of it or just before. If this is an issue, this should be addressed at proof reading (as the two columns layout will change the positioning of figures anyway).

*>Section 2.2 – first paragraph: To my knowledge MeteoIO isn't a numerical model but rather a pre- processor of input data. Clarification would be welcome.*

The mentions of MeteoIO have been rephrased and supported by a new reference as for a few years many users would rely on a provided example code to call MeteoIO standalone to prepare data files. For the last 18 months, this has been made more official by providing a proper executable for data preparation as part of the source tree (and it is now compiled by default). In this respect, MeteoIO's online documentation is obsolete when it refers to MeteoIO being "only" a library.

*>Section 3: As far as I can see both section 2 and 3 are implementation. A better separation of content would be welcome. Again referring to INI as the informal standard for configuration files has to be justified. In general I would claim that YAML is at least as widespread as INI, but there can be differences between communities that for legacy reasons stick with INI. References would help in this part.*

Sections 2 and 3 have been heavily reworked and several paragraphs have been moved to a different location. Regarding the INI standard, see our reply above. So far, we wanted to avoid being too specific by giving model names that rely on INI files in order to avoid being too "cryospheric sciences community" focused.

*>Section 3.2: I am sure I would agree on this being the architecture. As mentioned earlier, to simplify reading a general introduction followed by a proper problem description and analysis, then a section on methodology or approach that contains the architecture, leading up to the implementation and a discussion of this would simplify the reading of the paper. As sections 2 and 3 are somewhat interleaved and not clearly defined scope wise a restructuring would be welcome.*

The paper has been restructured accordingly.

*> Section 3.3: Not sure I understand what is meant by semantic names, seems repetitive.*

This has been replaced by "symbolic names". These names are not directly color names, but function names (such as "warning", "error"…) so this leads to more consistency and can easily be styled differently without having to change the colors everywhere.

*>Section 3.7: The work flow description would benefit from interaction or sequence diagrams. It is not clear if GUI and model has to run in the same environment or if remote control is possible, This*

*would be interesting to know. Also is this more than a configuration system? Is it also a execution environment?*

The supported execution environment has been described in the "Workflows" section: local execution by spawning a new process, no support for batch schedulers.

*> Section 3.8: I can't see that a numerical model (nor multiple) is loaded into INIShell. Rephrasing would be welcome to clarify what a numerical model is.*

This has been rephrased.

*>Section 4 – second paragraph: For the time being running numerical models is commonly done in HPC than cloud environments, but could depend on the community and type of model. Referring to the cloud without addressing HPC would be oversimplifying the complexity of numerical models. Also referring to my comment on section 3.7, a clarification of execution environment would be beneficial up front, not indirectly addressed in discussion.*

This has been rephrased as "directly on the web" for clarity. The supported execution environment has been described in the "Workflows" section: local execution by spawning a new process, no support for batch schedulers.

*>Section 4 – third paragraph: file formats for what?*

This has been clarified in the text (file formats for the configuration files, namely the INI files).

*>Section 5: I can't see that the statement that a GUI gives better control over the configuration than a text based solution. Justification would be required, also in the discussion. And the justification should also refer back to the problem description that is the foundation for the paper.*

This justification has been brought in the new section 1.3 "Graphical User Interfaces" with two references that are based on (technically minded) user surveys. The better control over the configuration is a consequence of the reduced learning curve and improved educational value (even for quite advanced users). The link to section 1.3 has explicitly been made.

---

## Author Response (AR3)

**Reply to the reviewers:**

> *Page 2*
> *"making a steep"*
> *--> making for a steep*

Done as suggested

> *"Open Source"*
> *Is this a proper name?*

Capitalization has been removed

> *Page 3*
> *"some varying"*
> *Grammar*

This has been rephrased

> *"the second category of users gets back to the WSL/SLF"*
> *--> requests support from?*

This has been rephrased as suggested

> *"don't have the same long learning curve"*
> *I would encourage avoiding contractions.please fix throughout*

We have removed all contractions in the manuscript

> *"and are more intuitive and educational than CLI"*
> *delete*

This has been replaced by the phrasing from the cited paper

> *"at deeper levels"*
> *delete*

Done as suggested

> *"a significant investment"*
> *Time, money, and code?*

This has been rephrased as suggested

> *"quite fast"*
> *--> quickly*

Done as suggested

> *"short term"*
> *--> short-term*

Done as suggested

> "editor"
> --> developer?

Done as suggested

> "even to keep the product running,"
> Unclear

This has been rephrased

> Page 4
> "standard toolkits"
> --> "Ui-toolkits"

This has been rephrased as suggested

> "languages"
> --> "programming languages"?

This has been rephrased as suggested

> "the product"
> What is the product in this case? GUI? Be clear

This has been rephrased

> "This practically means"
> --> "Practically this means"

This has been rephrased as suggested

> "when another intern or student with the proper skill set can be found and funded"
> I guess the academic mindset is implicit but maybe worth noting v.clearly

This has been rephrased for clarity

> "about the new options"
> That are added after the intern leaves?

This has been rephrased for clarity

> "in order to be useful to the end user."
> Abrupt ending. Need a linkage with what this paper is solving

This has been rephrased for a proper logical articulation with the section that follows

> "Requirements"
> Of what?

This has been rephrased

> *Page 5*
> *"among others."*
> *--> such as?*

Inishell has also been tested on Haiku and FreeBSD. It should also work on Android and iOS but this has not been tested so far.

> *"INI2"*
> *I'm not loving a Wikipedia link but I cannot find anything better to suggest.*

Although there are some web pages by Microsoft about INI files, they only cover application specific issues. The W3C also has some documentation about INI files but it is very rudimentary. Some individual developers' blogs expose some interesting ideas about INI files (such as why they are better suited to store configuration information than XML, YAML or JSON files) but are often loosely structured and don't contain the general, basic information. The Wikipedia entry is by far the most complete (and vendor neutral) generic documentation about the INI file format.

> *"It is"*
> *It = what?*

This has been rephrased

> *Page 6*
> *"principle of operation"*
> *Do you mean work flow or similar?. This is not clear to me*

This has been rephrased

> *"therefore"*
> *--> still*

Done

> *"The user workflow is described in Fig."*
> *This is a nice addition*

> *Page 8*
> *"The approach that has been laid out by (Havens et al., 2020) relies on Docker images"*
> *I am not sure this section adds anything, indeed it felt disjoint. Maybe worth moving to lit*
> *review?*

The last paragraph of this section has been reworked in order to flow more naturally and to better fit with the rest of the paper. The whole section has been moved to the literature review and adapted in order to fit there (so it is logically articulated with the sections before and after).

> *Page 9*
> *"his or her"*
> *--> their*

This has been consistently done throughout the manuscript

> *"inheriting the Atomic class"*
> *Which is what?*

This has been rephrased

> *Page 10*
> *"his or her"*
> *--> their*

This has been consistently done throughout the manuscript

> *"and merged back before writing out"*
> *I assume no conflicts really happen?*

No conflicts can happen because the INIParser only contains properties that could not be stored in the widgets themselves.

> *"his or her"*
> *--> their*

This has been consistently done throughout the manuscript

> *Page 11*
> *"(e. g."*
> *No Space*

Done as suggested

> *Page 13*
> *"Figure 8. Simple example of templates"*
> *Maybe expand this caption to be more descriptive. They should be stand alone. Check*
> *the other captions too*

All captions have been checked and expanded when necessary.

> *Page 16*
> *"as a fat client"*
> *native client?*

This has been rephrased as suggested

> *"very fast"*
> *--> quickly*

Done as suggested

> *"This means that end users don't need to"*
> *--> "do not"*
> *There are a few other contractions in this. Please avoid contractions.*

This has been consistently done throughout the manuscript